# Comparison of Efavirenz and Doravirine Developmental Toxicity in an Embryo Animal Model

**DOI:** 10.3390/ijms241411664

**Published:** 2023-07-19

**Authors:** Daniela Zizioli, Sara Ferretti, Giorgio Tiecco, Luca Mignani, Eugenio Monti, Francesco Castelli, Eugenia Quiros-Roldan, Isabella Zanella

**Affiliations:** 1Department of Molecular and Translational Medicine, University of Brescia, 25123 Brescia, Italy; daniela.zizioli@unibs.it (D.Z.); s.ferretti002@studenti.unibs.it (S.F.); luca.mignani1@unibs.it (L.M.); eugenio.monti@unibs.it (E.M.); isabella.zanella@unibs.it (I.Z.); 2Division of Infectious and Tropical Diseases, ASST Spedali Civili di Brescia, 25123 Brescia, Italy; g.tiecco@unibs.it (G.T.); francesco.castelli@unibs.it (F.C.); 3Department of Clinical and Experimental Sciences, University of Brescia, 25123 Brescia, Italy; 4Cytogenetics and Molecular Genetics Laboratory, Diagnostic Department, ASST Spedali Civili di Brescia, 25123 Brescia, Italy

**Keywords:** doravirine, efavirenz, non-nucleoside reverse transcriptase inhibitors, NNRTI, HIV, developmental safety, zebrafish embryo

## Abstract

In the past, one of the most widely used non-nucleoside reverse transcriptase inhibitors (NNRTI) in first-line antiretroviral therapy (ART) of HIV infection was efavirenz (EFV), which is already used as a cost-effective treatment in developing countries due to its efficacy, tolerability, and availability. However, EFV also demonstrates several adverse effects, like hepatotoxicity, altered lipid profile, neuropsychological symptoms, and behavioral effects in children after in utero exposure. In 2018, another NNRTI, doravirine (DOR), was approved due to its similar efficacy but better safety profile. Preclinical safety studies demonstrated that DOR is not genotoxic and exhibits no developmental toxicity or effects on fertility in rats. Zebrafish (*Danio rerio*) embryos have been widely accepted as a vertebrate model for pharmacological and developmental studies. We used zebrafish embryos as an in vivo model to investigate the developmental toxicity of DOR compared to EFV. After exposure of the embryos to the drugs from the gastrula stage up to different developmental stages (30 embryos for each arm, in three independent experiments), we assessed their survival, morphology, hatching rate, apoptosis in the developing head, locomotion behavior, vasculature development, and neutral lipid distribution. Overall, DOR showed a better safety profile than EFV in our model. Therapeutic and supra-therapeutic doses of DOR induced very low mortality [survival rates: 92, 90, 88, 88, and 81% at 1, 5, 10, 25, and 50 μM, respectively, at 24 h post fecundation (hpf), and 88, 85, 88, 89, and 75% at the same doses, respectively, at 48 hpf] and mild morphological alterations compared to EFV exposure also in the sub-therapeutic ranges (survival rates: 80, 77, 69, 63, and 44% at 1, 5, 10, 25, and 50 μM, respectively, at 24 hpf and 72, 70, 63, 52, and 0% at the same doses, respectively, at 48 hpf). Further, DOR only slightly affected the hatching rate at supra-therapeutic doses (97, 98, 96, 87, and 83% at 1, 5, 10, 25, and 50 μM, respectively, at 72 hpf), while EFV already strongly reduced hatching at sub-therapeutic doses (83, 49, 11, 0, and 0% at 1, 5, 10, 25, and 50 μM, respectively, at the same time endpoint). Both DOR at therapeutic doses and most severely EFV at sub-therapeutic doses enhanced apoptosis in the developing head during crucial phases of embryo neurodevelopment and perturbed the locomotor behavior. Furthermore, EFV strongly affected angiogenesis and disturbed neutral lipid homeostasis even at sub-therapeutic doses compared to DOR at therapeutic concentrations. Our findings in zebrafish embryos add further data confirming the higher safety of DOR with respect to EFV regarding embryo development, neurogenesis, angiogenesis, and lipid metabolism. Further studies are needed to explore the molecular mechanisms underlying the better pharmacological safety profile of DOR, and further human studies are required to confirm these results in the zebrafish animal model.

## 1. Introduction

Non-nucleoside reverse transcriptase inhibitors (NNRTIs) were first identified almost 30 years ago as in vitro selective inhibitors of HIV-1 replication. NNRTIs prevent human immunodeficiency virus 1 (HIV-1) replication by non-competitively inhibiting reverse transcription. NNRTIs bind to a unique allosteric pocket on HIV-1 reverse transcriptase (RT) and induce a conformational change in the substrate-binding site, interfering with RT-DNA polymerase activity. First-generation NNRTIs include nevirapine, delavirdine, and efavirenz (EFV). Second-generation NNRTIs, with a better resistance profile, include etravirine, approved in 2008, and rilpivirine, approved in 2011, respectively. In 2018, doravirine (DOR) was approved, being the most novel NNRTI, which, compared to the older NNRTIs, demonstrated a better safety profile with a similar efficacy [1,2,3].

EFV, a benzoxamine, was one of the most widely used NNRTIs in first-line antiretroviral therapy (ART) in the past. Nevertheless, and although currently no longer recommended for treatments by most of the international guidelines, it is still considered as an alternative cost-effective option in treatment guidelines for treating people with HIV-1 in developing countries due to its efficacy, tolerability, and availability regimens [4]. For a long time, EFV has been linked to fetal neurotoxicity and deficits in neurodevelopment and its use was then contraindicated during pregnancy [5,6,7]. Only after that no signs of EFV-related pregnancy complications or teratogenicity were reported in a meta-analysis including more than 2000 first-trimester exposures [8], EFV was admitted as part of the ART regimen in pregnant woman with HIV-1 [9].

Regarding DOR, preclinical safety studies demonstrated that it is not mutagenic nor genotoxic in vitro. No embryo-fetal developmental toxicity and no effects on the fertility of female and male rats have been observed [10]. Currently, DOR is administered to both antiretroviral-naïve and -experienced patients with HIV-1, although safety data for DOR mainly derived from two phase 3 clinical trials [2,3].

In recent years, zebrafish (*Danio rerio*) embryos have been widely accepted as a prominent vertebrate model for toxicological, pharmacological, and developmental studies in recent years due to their high fecundity, rapid organogenesis, short generation, and transparent bodies of embryos and larvae [11]. In this study, zebrafish embryos were used as an in vivo model to further investigate the developmental toxicity of DOR compared with EFV.

## 2. Results

### 2.1. Doravirine Has a Low Impact on Survival, While Efavirenz Induces a High Mortality Rate in Zebrafish Embryos at Human Therapeutic Doses

Chemical structures of DOR and EFV are shown in Figure 1. We first assessed whether DOR and EFV exert evident toxic effects in zebrafish embryos by first evaluating the embryo survival rates at 24 and 48 h post fertilization (hpf) after embryo direct drug exposure with the immersion method at the gastrula stage (4 hpf) [12]. Thirty embryos, for each condition and each of 3 experiments, were exposed to treatments. As expected, the survival rate was around 85–95% for the negative control embryos compared to lower than 10% in the positive control embryos at both time endpoints, respectively, according to the OECD TG 236 guidelines [12]. DOR exposure exhibited a mild effect on the survival rate up to the 25 μM dose and in a stronger effect at the highest dose (50 μM, about 25X of the human C_max_) at both time endpoints (with survival rates of 92, 90, 88, 88, and 81% at 1, 5, 10, 25, and 50 μM, respectively, at 24 h hpf, and 88, 85, 88, 89, and 75% at the same doses, respectively, at 48 hpf]. In contrast, survival rate of embryos treated with EFV was significatively reduced both at 24 and 48 hpf already at the lowest dose (1 μM, about 0.1X of the human EFV Cmax) (survival rates 80, 77, 69, 63 and 44% at 1, 5, 10, 25 and 50 μM respectively at 24 hpf and 72, 70, 63, 52 and 0% at the same doses respectively at 48 hpf) (Figure 1). The survival rates of DOR-treated embryos at a more advanced larval stage (144 hpf) was not evidently further decreased except for the highest 50 μM dose, suggesting a lower toxicity of DOR within and above the human therapeutic range (with survival rates of 90, 89, 88, 86, and 69% at 1, 5, 10, 25, and 50 μM, respectively (Appendix A).

### 2.2. Doravirine and Efavirenz Affect the Morphological Phenotype with Different Severities

In order to examine whether DOR or EFV induce developmental malformations, the morphology of the treated embryos was carefully evaluated at 24 and 72 hpf, respectively, through visual microscopic inspections considering changes in the head and tail morphology, the maintenance of a correct anterior–posterior (AP) axis or its deformation, normal growth or growth delay, the absence or presence of pericardial edema, and correct or altered somite formation (V-shaped or U-shaped somites). Embryos with morphological defects were identified and classified into embryos with a normal, mild, or severe phenotype based on the presence or absence of the above morphological endpoints (Table 1). A mild phenotype was defined as the presence of at least one of the mild abnormalities described in the table; conversely, a severe phenotype was considered in the presence of at least one of the severe malformations.

Percentages of malformed larvae (defined as having at least one malformation) were recorded at both endpoint times for each dose (Figure 2 and Figure 3B,C). No evident malformations were observed for most embryos compared to the control embryos after DOR exposure at both time endpoints and all tested doses, except for the mild abnormalities that were noted in a very small percentage of embryos from the doses of 5–10 μM (about 2–4× the human C_max_ for DOR) (0, 2, 3, 5, and 5% of embryos with mild abnormalities at 1, 5, 10, 25, and 50 μM at 24 hpf, respectively, and 1, 2, 5, 5, and 7% at the same doses at 72 hpf, respectively). We frequently observed a slightly curved tail in treated embryos in comparison with the control embryos at 24 hpf at all tested doses (Figure 2A), with a slightly smaller body length observed at 72 hpf at the highest 50 μM dose (Figure 3A). Both mild morphological alterations were suggestive of a mild growth delay of the DOR-exposed embryos. In contrast, EFV exposure led to grossly visible embryo malformations even with the lowest 1–5 μM doses (about 0.1–0.5× the human C_max_ for EFV) (68, 76, 80, 80, and 88% of embryos with at least one type of malformation at 1, 5, 10, 25, and 50 μM doses, respectively, at 24 hpf, and 70, 82, and 87% at 1, 5, and 10 μM doses, respectively, at 72 hpf, with no living embryos observed with the 25 and 50 μM doses at this time endpoint). We observed evident growth delays with dose-dependent and strong body length shortening, AP axis alteration, somite disruption, and tail necrosis in most embryos at 24 hpf (Figure 2A), and net worsening conditions at 72 hpf (Figure 3A), respectively, with the presence of moderate-to-grave cardiac edema, severe disorganization of the AP axis, shorter, detached, and curved tails, severe delay of growth with tail necrosis up to the 10 μM dose, and death of all the embryos at the 25 and 50 μM doses. All these morphological observations strongly suggested that DOR only mildly affected zebrafish embryo development, while EFV seemed to severely disrupt embryo growth and development already at sub-therapeutic doses.

### 2.3. Doravirine Only Slightly Affects the Zebrafish Embryo Hatching Rate at Supra-Therapeutic Doses while Efavirenz Strongly Reduced Hatching already at Sub-Therapeutic Doses

The hatching period in zebrafish embryos typically occurs between 48 and 72 hpf, respectively, and represents the transition point from a developing embryo to a free-living individual, the larva. This process is finely regulated by many endogenous and environmental factors and has been shown to be sensitive to a variety of chemical agents [13]. For hatching rate evaluation, 30 embryos for each condition and each of 3 experiments were exposed to treatments. No significant effects of DOR on hatching at 72 hpf were observed up to the 10 µM dose, while a slight but significant reduction in the hatching rate was observed at the doses of 25 and 50 µM, respectively (Figure 4A) (97, 98, 96, 87, and 83% of hatched embryos at 1, 5, 10, 25, and 50 μM DOR doses, respectively). Conversely, dose-dependent and significantly decreased hatching rates were observed in EFV-treated embryos already from the lowest 1 µM dose exposure and up to the 10 µM dose, which is roughly around the human therapeutic C_max_ dose (Figure 4B) (83, 49, 11, 0, and 0% of hatched embryos, corresponding to the 1, 5, 10, 25, and 50 µM EFV doses, respectively). As shown in the figure, no embryo treated with EFV survived at the highest 25 and 50 µM doses at this time during the hatching rate recording.

### 2.4. Doravirine and Efavirenz Enhance Apoptosis in the Zebrafish Embryo Head during the Crucial Phases of Neurodevelopment

In zebrafish embryos, physiological hatching requires embryo movement, which is strictly related to a proper neurodevelopment. Earlier, we described that while DOR did not evidently affect the hatching rate at the human therapeutic dose, EFV was able to either significantly reduce or delay hatching already under sub-therapeutic exposures. Extending from this, EFV has been shown to induce neuronal damage and apoptosis in both in vivo and in vitro models [14,15,16]. In order to verify whether DOR or EFV exposure result in the induction of apoptosis in the zebrafish embryo’s head during the most crucial phases of the development of the central nervous system, we stained treated embryos with acridine orange (AO). To test our hypothesis, we chose the lowest 1 and 5 μM doses for both drugs (due to the high mortality of the EFV-treated embryos observed at the highest doses) with thirty embryos for each condition and each of the three experiments, and performed AO staining experiments at 48 hpf, a crucial stage for neurodevelopment in zebrafish embryos during which the late stages of neurogenesis occur with the development of the definitive midbrain and hindbrain structures. We observed AO-stained embryos under a fluorescence microscope and also measured the total fluorescence in the homogenates of embryo heads. Compared to the control embryos, a significantly higher quantity of apoptotic cells (displayed as green, fluorescent spots) were observed in the brains of embryos treated with both drugs and at both doses (Figure 5A). Green, fluorescent spots were mainly localized to the anterior regions of the embryo head that correspond to the areas of the telencephalon and the diencephalon. The fluorescence intensity of the homogenized embryo heads was indeed higher than in the control embryos for both drugs and at both doses, with a very high fluorescence intensity observed for the 5 μM EFV-treated embryo heads (Figure 5B). These results suggest that both drugs, but EFV in particular, and both in a concentration-dependent manner, induced apoptosis in the heads of the zebrafish embryos during a crucial neurodevelopmental stage.

### 2.5. Doravirine and Efavirenz Treatments Perturbed the Swimming Behavior of the Zebrafish Larvae

In recent years, zebrafish embryos have been used to correlate behavioral patterns (locomotor activity and swimming velocity) with the effects of chemical compounds on brain development [17,18]. Swimming behavior is a result of the activity of the nervous system, which is necessary for fish to survive and reproduce in the wild. Considering the above results on the effects of DOR and EFV on the hatching rate and apoptosis in the developing head region, the dose-dependent neuropsychological symptoms associated with EFV treatment [19,20], the described association between the in utero exposure to EFV and the lower behavioral self-regulatory abilities and motor skills of exposed children [7], but the less frequent and milder adverse effects of DOR associated with the nervous system in humans [2,3], we decided to assess swimming behavior of our treated larvae. To this end, after continuous EFV or DOR exposure from 4 hpf to 144 hpf, respectively, the swimming behavior of twelve exposed embryos for each condition and in each of the three distinct experiments was analyzed with a light–dark locomotion test. Compared to the control embryos, the total (light + dark) distance swam by the larvae exposed to DOR and EFV and their movement speed showed evident alterations. In particular, we observed movement activation with longer and faster movements in the DOR-treated larvae up to the 5 μM dose, while reduced swimming in terms of both the distance traveled and the velocity were observed at the highest 10 and 25 μM doses, respectively (Figure 6). For EFV, we were able to only evaluate its sub-therapeutic doses (1 and 5 μM) due to the high embryo mortality that was previously observed with the higher doses. EFV exposure resulted in reduced movements (with a shorter distance and a lower speed) (Figure 6). These results suggest that DOR and EFV exert different effects on motor skills in the drug-exposed developing zebrafish.

### 2.6. Efavirenz Strongly Affects Zebrafish Embryo Angiogenesis Compared to Doravirine

Based on past research, EFV has also been shown to exhibit an anti-angiogenetic effect on the chick chorioallantoic membrane [21]. No data are available on the effects of DOR on the vascular compartment. Thus, we assessed and compared the effects of DOR and EFV at their lowest 1 and 5 μM doses on developmental angiogenesis, considering for this purpose the development of the intersomitic vessels (ISVs), the caudal venous plexus (CVP), and the formation of the sub-intestinal venous plexus (SIVP). For all analyses, thirty embryos, for each condition and each of the three experiments, were exposed to the treatments.

In zebrafish embryos, vasculature structures typically arise from 22 to 72 hpf, respectively. The formation of the ISVs are usually the first sign of angiogenesis, developing around 22 hpf between the somites, sprouting dorsally from the dorsal aorta and reaching the dorsal-most region of the somites to become part of the trunk vasculature [22]. The CVP is a dense capillary network comprising interconnecting venous tubes located in the tail of zebrafish embryos, starting at 25 hpf through the angiogenic sprouting of endothelial cells from the posterior cardinal vein and forming a mature venous vascular network at 48 hpf. The presence of intercapillary spaces in the CVP indicates a proper remodeling of the blood vessels [22]. The Tg (*kdrl*:EGFP) zebrafish line, characterized by the expression of enhanced green fluorescent protein (EGFP) in the vessels under the control of the vascular-endothelium-specific promoter of the zebrafish vascular endothelial growth factor receptor *kinase insert domain receptor like* (*kdrl*) gene, also known as *vascular endothelial growth factor receptor 2* (*vegfr-2*), is a powerful tool that is typically used to observe vascular growth in vivo, to study drug responses in live embryos, and to visualize the ISVs and CVP [23]. To assess the development of the ISVs and CVP under exposure to DOR or EFV, we used this transgenic line.

At 30 hpf, treatment of embryos with DOR did not evidently affect the correct formation of ISVs at the lowest 1 μM dose compared to control embryos, while we observed a slightly lower EGFP expression in ISVs but proper morphology at the 5 μM dose (Figure 7A). In contrast, EFV treatment at both doses caused a strong downregulation of EGFP expression and evident deformities of ISVs in survived embryos at both doses, suggesting a strong disruption of angiogenesis (Figure 7A). Moreover, EFV treatment at both doses caused a strong downregulation in EGFP expression and produced evident deformities of the ISVs in the survived embryos at both doses, suggesting a strong disruption of angiogenesis (Figure 7A). We also measured the length of the ISVs in embryos exposed to the drugs (ten embryos per group were analyzed in each of the three experiments) and found that, while DOR did not affect ISV length at the lowest dose, and slightly reduced it with a 5 μM concentration, EFV exposure significantly decreased the length of these structures in the survived embryos in comparison with the control embryos (Figure 7B,C).

Extending from this, as shown in Figure 8, DOR exposure reduced the number of CVP intercapillary spaces in a dose-dependent manner, while EFV exposure resulted in an even more evident reduction at the 1 μM dose, and in a clear enlargement of the CVP area with an evident disruption of its morphology at the 5 μM dose, respectively. Together, these results suggest that both drugs may exhibit anti-angiogenetic effects during zebrafish embryo vessel development, with EFV showing the highest disrupting impact even at sub-therapeutic doses.

SIVP development is frequently used in zebrafish embryo studies to screen chemicals with angiogenetic properties [24]. To further investigate the effects of DOR and EFV on angiogenesis, we explored their impacts on the development of the SIVP, an easily visible network of vessels that in zebrafish embryos is involved in the uptake of nutrients from the yolk. The SIVP develops in both the left and right dorsolateral sides of the embryo yolk, starting at about 30 hpf with the migration of cells from the posterior cardinal vein, and it is completely shaped at 72 hpf, with the complete formation of the intercapillary branches within the basket [18]. We exposed zebrafish embryos to both drugs (1 and 5 μM doses, respectively) or with fish water plus 0.1% DMSO as the control until 72 hpf following which the embryos were stained using the alkaline phosphatase (AP) assay to visualize the development of the SIVP. Treatment of the embryos with DOR evidently did not affect the SIVP formation at the lowest 1 μM dose compared to the control embryos, while we observed only a slightly smaller SIVP basket at the higher 5 μM dose (Figure 9A). In contrast, EFV treatment, at both doses and in a dose-dependent manner, caused a strong reduction in the size of the SIVP basket in the survived embryos, with almost no basket formation observed in some of them at the higher 5 μM dose (Figure 9A), further suggesting a strong EFV-induced disruption of the angiogenesis process, but also a significantly lower anti-angiogenetic effect of DOR. The number of the SIVP branches was then counted for both the treated and control embryos, and we observed a slight but significant reduction in the DOR-treated embryos, and a strong decrease in the number of sprout vessels present in the EFV-treated embryos. For both drugs, these effects were observed at both doses in a dose-dependent manner (Figure 9B).

### 2.7. Efavirenz, but Not Doravirine, Disturbs Neutral Lipid Distribution in the Zebrafish Embryos, while Both Drugs Altered the mRNA Expression of the Adipogenesis Master Regulator Genes srebf1 and pparg

DOR-based treatment regimens have previously been demonstrated to result in a better lipid profile in people with HIV-1, although accompanied with a greater weight increase in comparison with the EFV-based regimens [25,26,27]. We thus monitored neutral lipids by staining the treated embryos with Oil Red O (ORO) at 144 hpf. Until 120 hpf, the zebrafish embryos are not able to actively feed and rely on the consumption of nutrients that have been accumulated in the yolk, mainly consisting of lipids. Following this, neutral lipids are mainly visible in the yolk, head, heart, vasculature, and swim bladder in this developmental phase. At 144 hpf, the yolk is nearly completely consumed, and the head, heart, and vasculature only show negligible levels of stainable neutral lipids [28]. At 144 hpf, neutral lipid staining appeared to be similarly distributed in the control and DOR-treated embryos (both doses), with predominant staining observed of the swim bladder, and residue staining of the yolk and of the circulating lipids in the heart and vasculature. For ORO staining, thirty embryos for each condition and each of the three experiments were exposed to treatments. In embryos treated with at 5 μM DOR dose, a little area of pericardial edema was also visible in a small percentage of embryos (8%). Conversely, EFV-treated embryos showed lesser staining of the swim bladder (or a smaller swim bladder) and vasculature at both exposure doses. For both exposures at this time endpoint, the embryos showed pericardial edema. Further, embryos treated with the 5 μM EFV concentration revealed a high lipid storage in the liver area (Figure 10A), suggestive of hepatic steatosis.

We also evaluated the mRNA expression of the crucial transcription factor genes involved in adipogenesis sterol regulatory element-binding transcription factor 1 (srebf1), peroxisome proliferator-activated receptor gamma (pparg) and CCAAT enhancer-binding protein alpha (cebpa). For these experiments, we chose the same drug doses (1 and 5 μM) but a different time endpoint (48 hpf), considering that what we observed at 144 hpf should be the result of the previous action of these genes. We observed a trend of decreased expression in the adipogenesis master regulator genes srebf1 and pparg for both doses and both drugs, which was significant for EFV at the highest dose, while cebpa expression was unmodified by both drugs (Figure 10B).

## 3. Discussion

Our results showed that DOR had minimal effects on embryogenesis with a slight decrease observed in the survival rates, but neither gross phenotypic defects during neurogenesis nor lipid accumulation in the zebrafish embryo model up to 25× the therapeutic concentration were observed, contrarily to EFV. Notwithstanding, improper angiogenesis, increased apoptosis in the head region, and behavior alterations were observed in both the EFV- and DOR-treated embryos. A summary of our observations is shown in Table 2.

Due to its good safety and efficacy profile, EFV was preferred as the first-line agent for treating patients with HIV-1 for a long time until the first HIV integrase inhibitor (INSTI) drugs were registered. The safety profile of EFV has been well-characterized and mainly comprises dermatologic, hepatic, lipidemic, and central nervous system manifestations [29]. Neuropsychiatric adverse events have been described with a rate as high as 25–70% (which include abnormal dreams, insomnia, somnolence, hallucinations, dizziness, impaired concentration, aggressive behavior, severe depression, and suicidal thoughts) [30,31,32]. The manifestation of neuropsychological symptoms have been associated with the dose of EFV, with improvements having been reported following dose reduction [19]. An association between the in utero exposure to EFV in children with poorer language abilities, emotional/behavioral self-regulatory capacities, and gross and fine motor skills has also been described [7]. Other adverse effects of EFV in people living with HIV include serum liver enzyme increase (reported in 2–7% of patients) and high-density lipoprotein (HDL)-cholesterol, total cholesterol, and triglyceride increase.

Concerns regarding the safety of EFV often lead in a switch to alternative EFV-free regimens in people living with HIV, both with drugs from other classes of antiretroviral drugs along with drugs from the same class (NNRTIs) [33]. In recent years, the relevant guidelines have recommended making the transition from NNRTIs to INSTI-based regimens based on their tolerability and potency compared with the NNRTIs.

DOR, the newest NNRTI, seeks to restore the utility of this class as a potential therapeutic option in the treatment of both naïve and experienced patients; nonetheless, few data on DOR safety are available in people living with HIV and scarce data have been published incorporating in vitro or in vivo animal models [10]. A phase IIb clinical trial that compared the efficacy and safety of DOR and EFV demonstrated that both drugs exhibit a similar level of antiviral activity. In general, DOR demonstrated a better neurological safety profile than the comparator, except for insomnia, which was more frequent in patients in the DOR group than in the EFV group (7.4% vs. 2.8%, respectively). More than one neurological event was in fact described in 26.9% of patients in the DOR group vs. 47.2% of patients in the EFV group, respectively (difference—20.4 [95% CI − 32.6, −7.5). Dizziness, abnormal dreams, and nightmares were the driven neurological adverse events reported in the EFV group [26].

The safety of DOR in people living with HIV was also assessed in two phase 3 clinical trials with most (77%) side effects considered of mild severity (grade 1) [2,3]. Adverse effects in the nervous system were described as mild and infrequent: headaches, dizziness, and somnolence were reported in 1–10% of patients, and disturbances in attention, memory impairment, paresthesia, hypertonia, and poor-quality sleep were reported in 0.1–1% of patients, respectively. Moreover, increased serum lipase was described in 10%, increased serum aspartate transaminase (AST) in 7%, and increased alanine transaminase (ALT) in 6% of patients, respectively [34].

Therefore, in clinical trials, DOR-based regimens have demonstrated to exert significantly fewer neuropsychiatric events except for insomnia and have a superior lipid profile compared to the EFV-based regimens [25,26,27]; although, a greater weight increase has been reported with DOR than with EFV at 48 but not at 96 weeks of treatment [35]. The newest NNRTI DOR resulted in the reduction in mean plasmatic low-density lipoprotein (LDL) cholesterol, HDL cholesterol, total cholesterol, and triglycerides compared with the increases observed in patients treated with EFV. Hypertriglyceridemia (500–1000 mg/dL) has been reported as a rare side effect in patients treated with DOR in the same trials (accounting for 0.3–0.6% of patients). Very few post marketing or real-life studies on DOR safety have been published, with most of them showing a favorable metabolic impact of the DOR-based regimens with improvements in the cardiovascular risk parameters [36,37,38,39]. Hepatic safety was also confirmed in these real-life studies [38].

The specific mechanisms underlying the adverse effects of EFV are not yet fully understood, and they have been extensively studied in both in vitro and in vivo models showing an extensive toxicity pattern in several cells, including neurons, hepatocytes, and tumoral cells. Neuropsychological adverse effects have been explained through the direct interference of EFV with neurotransmitters. EFV acts as a serotonin (5-HT)-6 receptor inverse agonist, as a 5-HT2A, 5-HT2C, and 5-HT3A receptor antagonist, and a blocker of the 5-HT transporter (5-HTT) [20,40,41]. The 5-HT2, 5-HT3, and 5-HT6 receptors are involved in cell division, differentiation, survival, and in the neuronal migration pathway, and EFV, as a 5-HT receptor (ant)agonist, could directly interfere with these processes, consequently leading to neuronal cell death. Moreover, EFV induces neuronal damage dose-dependently in vitro [14,15]. EFV-exposed rats showed delayed maturation (delayed eyelid opening), delayed reflex and motor development, and reduced body weights throughout the course of their lives [20]. EFV exposure also causes structural changes to the motor cortical layers, with reductions in the number of mature neurons observed probably due to neuronal apoptosis [20]. Acute treatment with EFV induced an anxiogenic-like effect, while sub-chronic treatment induced both anxiogenic-like and depressive-like behavior, with both effects having been found to be dose-related in mice. Finally, EFV was proven to possess a broader anti-seizure profile, probably by virtue of its pharmacological poly-target signature [42,43]. The cellular toxicity of EFV in human hepatic cells seems to be mediated by a decreased mitochondrial membrane potential, increased cellular superoxide production, oxidative injury, and induced apoptosis [44]. Jin and colleagues [16] also described increased apoptosis with decreased neural stem cell proliferation both in vitro and in mice induced by EFV exposure. EFV also seems to suppress angiogenesis in chick chorioallantoic membranes in vitro [21]. Interestingly, the cellular toxicity of EFV has been demonstrated across several cancer cells, which suggested an antitumor effect of the drug [45,46,47,48]. This effect has mainly been attributed to oxidative injuries, the induction of apoptosis or other forms of cell death, loss of nuclear integrity, and the induction of the DNA damage response pathway. Further, EFV significantly increased plasma cholesterol levels and elicited hepatic steatosis in mice: in vitro and in vivo in-depth analysis of the mechanisms behind these effects showed that EFV is a potent pregnane X receptor (PXR)-selective agonist in the liver [49].

Our results in the zebrafish embryo model showed that EFV induces high mortality, delayed maturation, severe malformations during development, delayed hatching, increased apoptosis in the brain region, altered locomotor behavior, a clear anti-angiogenic effect, and increased lipid accumulation in the liver just from its sub therapeutic doses alone.

As already mentioned, in utero EFV exposure has been linked to microcephaly and neurodevelopmental deficits [7,50]. We observed a dose-related delayed hatching in zebrafish embryos that could be associated with a decreased embryo motility induced by the drug along with a reduced distance travelled in a dose-related response that could be interpreted as a depressive-like behavior, as previously described in rats in which it was related to the increase in the striatal levels of 5-HT, dopamine, and noradrenaline [51]. As the hatching, movement, and swimming behavior of zebrafish embryos are also a natural reflection of the coordination ability of the body and integrative neuronal function, our data on embryo locomotor behavior could also reflect direct neural damage. In this respect, the enhanced apoptosis observed in the head regions of the EFV-treated embryos could be one of the mechanisms at the basis of EFV neurotoxicity and neurodevelopmental effects. These phenomena were in general agreement with those observed on perinatal mice exposed to EFV [20]. Additionally, our findings further confirm previous findings regarding EFV neurotoxicity by the use of a zebrafish embryo model.

To the best of our knowledge, no information regarding DOR effects in in vitro cellular models or in vivo animal models has been published to date, except for the preclinical safety studies released by the Food and Drug Administration [10], which demonstrated no mutagenic nor genotoxic effects in vitro and no embryo-fetal developmental toxicity with no effects on the fertility of female and male rats. A further study has been recently published in which damaging effects on pancreatic beta-cell functions were observed for EFV and rilpivirine among the NNRTIs, but not for DOR [52]. This is the first further study in which DOR toxicity has been investigated, using zebrafish embryos as a widely accepted vertebrate model for toxicological, pharmacological, and developmental studies [11].

In our study, DOR neither induced zebrafish embryo mortality even at its highest dose, nor caused gross phenotypic malformations: embryo growth looked normal in contrast with the EFV-treated embryos, excluding a slight reduction in the body length at the highest supra-therapeutic dose. Moreover, the hatching rate was not modified by DOR exposure if not at its very high and supra-therapeutic doses. However, the swimming behavior of DOR-treated embryos showed abnormalities with respect to the control embryos. DOR induced an increased locomotor activity (total distance travelled and velocity) at low-to-moderate concentrations (up to 5 μM) and decreased activity at higher doses (10 and 25 μM). Notwithstanding, significantly enhanced apoptosis in the head regions during a crucial phase of neurodevelopment was observed with both EFV and DOR exposures at all dosages that were assessed, although with a lower intensity in the DOR-treated embryos compared to the EFV-treated ones. In contrast, no effects on cell viability, apoptosis, or necrosis were observed after DOR exposure in a rat insulinoma cell line [52]. Our results suggest that, although significantly fewer neuropsychiatric events have been described in the DOR- than the EFV-treated patients, DOR exposure may also be related to a certain degree of neurotoxicity, which should be particularly considered during pregnancy exposure, but may also be responsible for the higher proportion of patients with insomnia compared to the patients treated with EFV, possibly suggesting a different mechanism of neurotoxicity for these two drugs. Further studies are needed to deepen the mechanisms of DOR neurodevelopmental toxicity and the possible role of apoptosis in further cellular models, especially of neurons and other brain cells.

Interestingly, both drugs seem to alter angiogenesis, an effect that has been previously described with EFV exposure [21], but for the first time here also evidenced with DOR exposure. The anti-angiogenic effect was particularly evident and dose dependent after EFV exposure even at the sub-therapeutic doses, with a significantly decreased activation of the *kdr* promoter and a significant reduction and deformation of the ISVs and the CVP intercapillary spaces. The disruption of angiogenesis was clearly less evident after DOR exposure, with a slight reduction in *kdr* activation, ISV length, and number of CVP intercapillary spaces. Further, EFV treatment caused a dramatic reduction in the size of the SIVP basket and of the number of its branches, while DOR exposure resulted in a slight effect only at the higher dose, suggesting that these two drugs may act on developmental angiogenesis through different mechanisms. To our knowledge, the effects of DOR angiogenesis have not been explored until now. More studies are therefore necessary to understand the mechanisms behind its partial disruption.

Finally, in our model of EFV exposure we observed decreased neutral lipids in the swim bladder and in the vasculature with their accumulation in the liver. While the reduction in these circulating lipids may also be determined by the observed disruption of vasculature induced by EFV, our results further suggest that EFV may act on adipogenesis and lipogenesis and may result in hepatic steatosis, as already observed in mice [49]. Studies on people living with HIV have shown that EFV is able to alter lipogenesis resulting in central adiposity with hepatic steatosis, peripheral lipoatrophy, and dyslipidemia [53]. Conversely, in this study, no accumulation of lipids was observed in the embryo liver until 144 hpf after DOR exposure, suggesting that DOR, at therapeutic concentrations, would not likely exert profound deleterious effects on lipid metabolism in the liver, in contrast to the effects exerted by EFV [49]. Also, we did not observe further gross alterations in neutral lipid distribution in the DOR-treated embryos, in agreement with the previous observation in that in clinical trials, DOR-based regimens have demonstrated to possess a superior lipid profile compared with the EFV-based regimens [25,26,27], although greater weight increase at 48 weeks after DOR treatment has been observed [36,37,38,39]. Currently, there are several ongoing trials that have been aimed to verify whether people living with HIV who had a significant weight gain after starting an INSTI regimen could improve their metabolic and cardiovascular health (loss weight, waist circumference, and fat and lean mass body composition) within about 1 year if they switch to a regimen containing DOR [54]. Nonetheless, very few studies on DOR safety have been published [36,37,38,39], but most of them show a favorable metabolic impact of these DOR-based regimens. Our results further confirm those observations in humans. Adipogenesis and lipogenesis are regulated by a complex transcriptional network comprising several transcription factors, the master regulators being peroxisome proliferator-activator receptor γ (PPARγ), CCAAT/enhancer-binding protein alpha (CEBP-α), and sterol regulatory element-binding transcription factor 1/sterol regulatory element-binding protein 1c (SREBF1/SREBP1c). Adipocyte differentiation is mediated by a finely orchestrated expression of adipogenic transcription factors, like PPARγ and CEBPα, and the subsequent lipid storage induced by SREBP1c, a lipogenic transcription factor. The alteration in the gene expression of these factors in relation with obesity and the associated metabolic abnormalities has been extensively in humans and animal models [55,56]. The lipid alterations found in the patients treated with EFV seems to be mediated by the inhibition of the SREBF1-dependent lipogenic and PPARγ- and CEBP-α-dependent adipogenic pathways as observed in the adipocyte cellular models [57,58,59]. The current study also measured the mRNA expression of the *srebf1*, *pparg,* and *cebpa* genes in the zebrafish embryos treated with EFV and DOR. As previously described in the in vitro human adipocyte models, we found that EFV causes a concentration-dependent suppression of the *srebf1* lipogenic transcription factor and a trend to a decreased expression of the *pparg* master regulator of adipogenesis, with no modulation of the *cebpa* gene expression. We observed similar effects (although with no significantly different values in comparison with the control embryos) after DOR exposure, suggesting that also this drug may exert a less important but to be considered effect on adipogenesis and lipogenesis, possibly through the same pathways altered by EFV exposure. Further studies are needed to deepen the effects of DOR on lipid metabolism and their biochemical and molecular mechanisms.

We recognize that the main limitation of our work is that we used an animal model to study and compare the safety profiles of DOR and EFV. Thus, further human studies and surveillance registries, as well as further in vitro and in vivo studies in cellular and animal models are necessary to confirm these results. The second limitation is that we did not thoroughly deepen the mechanisms responsible for our observations; although this is the first study exploring new toxic profiles of DOR. Nonetheless, our data on apoptosis in regions of the brain and angiogenesis inhibition may suggest the direction of future studies. Despite the above limitations, we feel that our multi-dimensional exploration of the toxic effects of these drugs may provide interesting clues for further investigation in humans.

Taken together, our results reveal that DOR is quite safe in zebrafish embryo development even at high supratherapeutic doses in contrast with EFV, which appears to be toxic and teratogenic even at very low subtherapeutic doses. However, swimming behavior, head region apoptosis, angiogenesis, and possibly the adipogenesis and lipogenesis pathways seemed to be partially affected during embryo development even after DOR exposure, although without grossly visible morphological malformations (except for a mild growth delay, indicated by a more curved tail at 24 hpf, and a smaller body length at 72 hpf in comparison with the control embryos, respectively), suggesting neurotoxic and antiangiogenetic potential effects of the drug. DOR effects on neurodevelopment, angiogenesis, and adipogenesis merit further investigations to understand their underlying mechanisms.

In conclusion, our findings in zebrafish embryos add further data confirming that the newest NNRTI DOR is safer than EFV with respect to embryogenesis, neurodevelopment, angiogenesis, and lipid accumulation. Nonetheless, our findings were observed in an animal model, meaning further surveillance and post-marketing human studies are warranted to confirm their translation to humans. The use of DOR-containing regimens needs to therefore be carefully balanced given the currently lesser clinical experience compared to other similar drugs, and the lack of data on the adverse effects associated with its long-term use.

## 4. Materials and Methods

### 4.1. Ethics Statement

The experiments took place at the Zebrafish Facility, Department of Molecular and Translational Medicine, University of Brescia, Italy. All animal experiments were conducted in accordance with the Italian and European regulations on animal care and the standard rules defined by the Local Committee for Animal Health (Organismo per il Benessere Animale) and were authorized by the Italian Ministry of Health (Authorization Number 585/2018).

### 4.2. Zebrafish Maintenance and Collection of Eggs

Zebrafish embryos were collected from the AB wild-type line and the transgenic line Tg (*kdrl*:EGFP), in which the zebrafish vascular endothelial growth factor receptor *kinase insert domain receptor-like (kdrl)* promoter drives the expression of enhanced green fluorescent protein (EGFP) in the blood vessels throughout embryogenesis. Animals were bred in a recirculating aquaculture system (Techniplast ZebTEC, Buguggiate, VA, Italy) in fish water (0.1 g/L Instant Ocean Sea Salts, 0.1 g/L sodium bicarbonate, 0.19 g/L calcium sulfate) at 28.5 °C in a 14 h light and 10 h dark daily cycle, as previously described [60]. Adult male and female animals were mated in the breeding box overnight. The next morning, freshly spawned eggs were collected, washed with fresh fish water, maintained at 28 °C in petri dishes containing fresh fish water until the onset of gastrulation (about 4 h post fertilization, hpf), and finally exposed to the drugs. Embryo staging was performed as described by Kimmel and colleagues [61].

### 4.3. Drug Exposure of Embryos

Stock solutions of DOR or EFV were prepared for embryo exposure by dissolving the drugs (Selleck Chemicals, Houston, TX, USA) in distilled water plus dimethyl sulfoxide (DMSO, 10% final concentration) (Sigma-Aldrich, Saint Louis, MO, USA). Exposure solutions at each tested concentration were freshly prepared by serial dilutions of the stock solutions (1 mM of active substance) in fish water with 0.1% final DMSO concentration. Alive embryos at the gastrula stage (4 hpf) were dechorionated to maximize the drug uptake, transferred to petri dishes, and exposed to DOR or EFV at the selected doses in the 1–50 µM range (1–5–10–25–50 µM) through the classic immersion method [62] by up to 24, 30, 48, 72, or 144 hpf, respectively, depending on the type of experiment. Only for the hatching rate were non-dechorionated eggs used. The dose range was selected on the basis of the human C_max_ of both drugs (2.2 μM and 9.1 μM for DOR and EFV, respectively) and considering higher doses up to a large multiple (25×) of the human C_max_ for the drug that was the main focus of this work (DOR). As negative control, embryos were exposed to 0.1% DMSO in fish water (expected mortality rate <10%). As a positive control for the survival rate experiments, we used 3,4-dichloroaniline (DCA) (Sigma-Aldrich) dissolved in fish water at the concentration of 3.74 mg/L (expected mortality rate >85–90%) [12,63]. For each type of analysis, 30 embryos for each arm in each of the 3 independent experiments were used.

### 4.4. Evaluation of Mortality, Gross Morphology, and Hatching Rate

Drug-induced mortality and embryotoxicity were essentially evaluated with the fish embryo toxicity test (FET) as previously described [60]. The survival rate was recorded at 24 and 48 hpf, respectively. and a dose-response graph was plotted. Morphology was carefully evaluated at 24 and 72 hpf, respectively, through visual inspections of anesthetized embryos (0.4% Tricaine) (Sigma-Aldrich) from head to tail under microscope direct visualizations with the Zeiss Axiozoom V13 microscope (Carl Zeiss AG, Oberkochen, Germany), equipped with a PlanNeoFluar Z 1x/0.25 FWD 56 mm lens and analyzed with Zen 3.5 (Blue version) (Carl Zeiss AG) (magnification 10× and 20×). As developmental indicators, we considered gross changes like alterations in the head and tail morphology, maintenance of a correct anterior–posterior (AP) axis or its deformation, normal growth or growth delay, absence or presence of pericardial edema, correct or altered formation, and the morphology of somites by grading their phenotype as normal, mild, or severe (Table 1). Percentages of malformed larvae were recorded at both endpoint times for each dose. Hatching rates were recorded at 72 hpf and plotted as a percentage of hatched larvae.

### 4.5. Acridine Orange Staining

Apoptotic cells were visualized in live embryos by acridine orange (AO) staining as previously described [64]. Briefly, at 48 hpf the control and treated embryos were dechorionated and incubated in fish water containing AO staining solution (10 mg/L) (Sigma Aldrich) for 20 min at room temperature. Larvae were then thoroughly washed with fish water, anesthetized with 0.4% Tricaine, quickly imaged under the fluorescent microscope (Zeiss Axio Zoom.V16 equipped with Zeiss Axiocam 506 color digital camera, Carl Zeiss AG), and analyzed using Zen 3.5 (blue Version) software (Carl Zeiss AG). Pools of 20 embryos for each treatment were then dissected and their heads were isolated and lyzed in 100% ethanol. Fluorescence was measured at an excitation wavelength (λ) of 490 nm and an emission λ of 525 nm with a microplate reader (Ensight Perkin Elmer, Waltham, MA, USA).

### 4.6. Behavior Assessment with the Light–Dark Locomotion Test

Embryos were exposed from 4 hpf to 144 hpf to DOR (1–5-10–25 μM), EFV (1–5 μM), or fish water plus 0.1% DMSO as the control. Experiments were performed essentially as previously described [60]. Briefly, for each treatment, 12 survived larvae at 144 hpf were collected in a 96 square well plate with one larva per well in a volume of 200 µL. The 96 square well plate was then put in the observation chamber of the *Danio Vision* Noldus system holder (Noldus, Wageningen, The Netherland), in an isolated noise-free room. The larvae were allowed to adapt for 30 min before video recording. The system was then set up to track movements (moved distance in 2 min time bins) for 2 h by 6 cycles of alternating light and dark 10 min periods. Data were analyzed using the Noldus *Ethovision* software (Noldus, https://www.noldus.com/ accessed on 21 June 2023). Movements were reported as the total distance (cm) travelled by the larvae and speed as mm/s, calculated under both light and dark stimuli.

### 4.7. Image Analysis of the Intersomitic Vessels (ISVs) and the Caudal Venous Plexus (CVP)

Blood vessels were visualized in live transgenic Tg (*kdrl*:EGFP) embryos under two different developmental stages: at 30 hpf for the development of the intersomitic vessels (ISVs), the first sign of angiogenesis in zebrafish embryos, and at 48 hpf for the formation of the caudal venous plexus (CVP), a network of interconnecting tubules that supports the blood flow in the tail during the early developmental stages.

At 30 hpf, embryos were anesthetized with 0.4% Tricaine and mounted in 1% low melting agarose gel. Fluorescent images depicting the vascular tree were acquired for the control and treated groups. The images were taken in a lateral position at a 10× and 20× magnification with the Zeiss Axiozoom V13 fluorescence microscope (Carl Zeiss AG) equipped with a PlanNeoFluar Z 1×/0.25 FWD 56 mm lens and were then processed with Zen 3.5 (Blue version) (Carl Zeiss AG). After the images were processed, the length of the ISVs was measured using ImageJ Fiji software 1.8.0 (Rasband, W.S., ImageJ, U. S. National Institutes of Health, Rockville, Bethesda, MD, USA) as previously described [65]. Two landmarks—one dorsal and one ventral—were established to define the section of the ISVs to be measured: ventrally, the commencement of the ISV at the upper border of dorsal aorta; dorsally, the bifurcation of the ISVs into the dorsal longitudinal anastomotic vessels (DLAVs). To measure the length of the ISVs, 10 embryos per group were analyzed. For each embryo, 4 intersegmental vessels corresponding to the yolk sac extension were selected. At 48 hpf, the embryos were then treated and visualized as above for fluorescent image captures of the CVPs.

### 4.8. Alkaline Phosphatase (AP) Staining of the Sub-Intestinal Venous Plexus (SIVP)

An alkaline phosphatase (AP) assay was performed to visualize the ectopic sprout formation of the sub-intestinal venous plexus (SIVP) in zebrafish embryos, as according to Serbedzija and colleagues [66]. The AP assay was performed as described elsewhere [24]. Briefly, at 72 hpf, the control and treated embryos were fixed in 4% paraformaldehyde (Sigma-Aldrich), put into 100% methanol, equilibrated in Tris buffer (comprising 100 mM Tris-HCl pH 9.5, 50 mM MgCl₂, 100 mM NaCl, and 0.1% Tween-20), and stained with nitro blue tetrazolium chloride (NBT) and 5-bromo-4-chloro-3′-indolyphosphate p-toluidine salt (BCIP) solution (Blue staining solution, Merk KGaA, Darmstadt, Germany). The images of the stained SIVP were taken in a lateral position at 32X magnification with a Zeiss Axiozoom V13 (Carl Zeiss AG) microscope, equipped with a PlanNeoFluar Z 1×/0.25 FWD 56 mm lens and Zen Pro software (Carl Zeiss AG). The number of SIVP branches was manually counted in all experimental groups according to Goi and Childs [67], and as previously described [68], and was then graphed using GraphPad Prism 8 (GraphPad Software, Boston, MA, USA).

### 4.9. Neutral Lipid Oil Red O (ORO) Staining

Neutral lipids were visualized by Oil Red O (ORO) staining (Sigma-Aldrich), according to the protocol of Yoganantharjah et al. [69]. At 144 hpf, the control and treated embryos were anesthetized with 0.4% Tricaine, washed and soaked in fish water, fixed in 4% paraformaldehyde for 12 h at 4°C, and washed three times in phosphate-buffered saline (PBS) 1X solution. The fish were then preincubated in 60% isopropanol in PBS 1× solution for 30 min, dyed with fresh 0.3% ORO (Sigma Aldrich) for 3 h, and thoroughly washed with 60% isopropanol in PBS 1X solution to minimize background. The stained embryos were then observed, and images were obtained with a Zeiss Axiozoom V13 (Carl Zeiss AG) microscope, equipped with a PlanNeoFluor Z 1×/0.25 FWD 56 mm lens at 20× magnification.

### 4.10. Quantitative Analysis of Gene Expression using Real-Time RT-PCR

Embryos were exposed from 4 hpf to 48 hpf to either DOR or EFV (1–5 μM), or fish water plus 0.1% DMSO as the control. Total RNA was extracted using the TRIzol reagent (Thermo Fisher Scientific, Waltham, MA, USA) according to the manufacturer’s protocol and was then quantified using the mySPEC micro-volume spectrophotometer (VWR International, Philadelphia, PA, USA). The RNA for each sample was then treated with RQ1-DNaseI (Promega, Madison, WI, USA) and 1 µg for each sample was reversed-transcribed using the Improm II Reverse Transcriptase (Promega) and oligo (dT) primers, following the manufacturer’s protocol. For quantitative RT-PCR, 20 ng of single-stranded cDNA was mixed with 2 µM of each primer and 5 µL of SYBR Green Master Mix (Biorad, Hercules, CA, USA) in a final volume of 10 µL in triplicate for each sample. RT-PCR was performed using the Viia7 system (Thermo Fisher Scientific, Waltham, MA, USA). The amplification profile was as follows: initial denaturation step at 95 °C for 1 min, followed by 40 cycles of two step amplification (95 °C for 15 s and 60 °C for 30 s), and a melting cycle. The calculation of the threshold cycle (Ct) values was performed using SDS 2.2 software (Applied Biosystems, Waltham, MA, USA) after automatically setting the baseline and the threshold. Each reaction was performed in triplicate and the relative expression of each gene was calculated with the 2^−∆∆Ct^ method, using *rpl13a* as a reference gene, as previously described [70]. Expression quantification was performed for *peroxisome proliferator-activated receptor gamma* (*pparg*), *sterol regulatory element-binding transcription factor 1(srebf1), and CCAAT enhancer-binding protein alpha* (*cebpa*), with *ribosomal protein L13a* (*rpl13a*) used as the housekeeping gene. Sequences of the specific primers used in this paper are available in Appendix A.

### 4.11. Statistical Analysis

Each experiment was repeated at least three times for both the control and the treated groups, with 30 embryos for each group and each experiment. Only for behavioral experiments were the number of embryos used varied: locomotion was analyzed in 12 larvae for each treatment and each experiment, and for ISV length, measures for 10 embryos for each treatment and each experiment were obtained. Data were presented as mean ± SD of all experiments. All graphs were plotted and analyzed using GraphPad Prism 8 (GraphPad Software). Significance was analyzed using the Student’s *t*-test; *p*-values less than 0.05 were considered as statistically significant (* *p* < 0.05; ** *p* < 0.005; *** *p* < 0.001; **** *p* < 0.0001). All laboratory assessments were performed by blinded laboratory personnel.

## Figures and Tables

**Figure 1 ijms-24-11664-f001:**
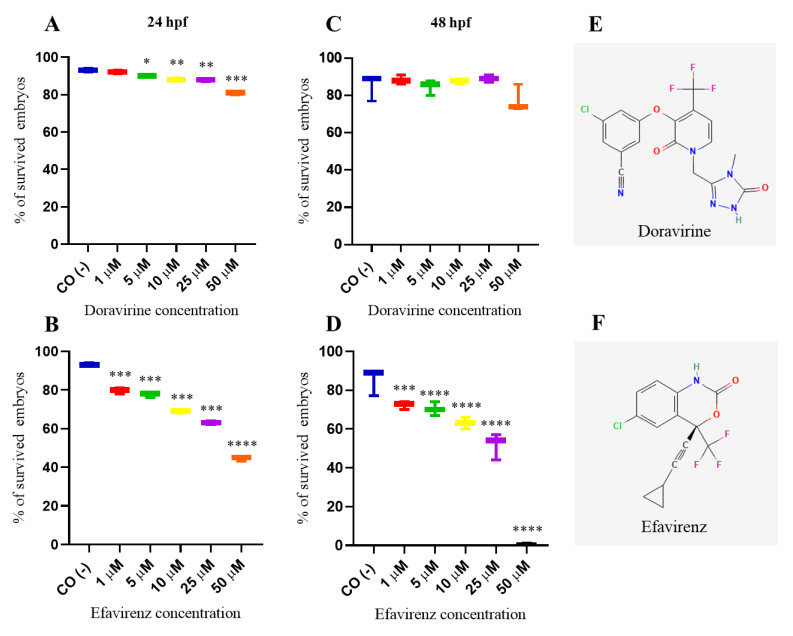
Survival rates of dechorionated zebrafish embryos at 24 (**A**,**B**) and 48 (**C**,**D**) hours post fertilization (hpf) after exposure to doravirine (DOR) or efavirenz (EFV) (1, 5, 10, 25, and 50 μM) at gastrula stage (4 hpf) with the immersion method. CO (-) represents the embryos exposed to the drug solvent only (fish water plus 0.1% dimethyl sulfoxide, DMSO). The *X*-axis shows drug doses used for the exposure of embryos; the *Y*-axis shows the corresponding survival percentages. Results are expressed as mean ± SD of 3 independent experiments, with 30 embryos for each experiment and each treatment. (* *p* < 0.05 vs. control group; ** *p* < 0.005 vs. control group; *** *p* < 0.001 vs. control group; **** *p* < 0.0001 vs. control group). (**E**,**F**) Two-dimensional structures of DOR, 3-chloro-5-[1-[(4-methyl-5-oxo-1*H*-1,2,4-triazol-3-yl)methyl]-2-oxo-4-(trifluoromethyl)yridine-3-yl]oxybenzonitrile, and EFV, (4*S*)-6-chloro-4-(2-cyclopropylethynyl)-4-(trifluoromethyl)-1*H*-3,1-benzoxazin-2-one.

**Figure 2 ijms-24-11664-f002:**
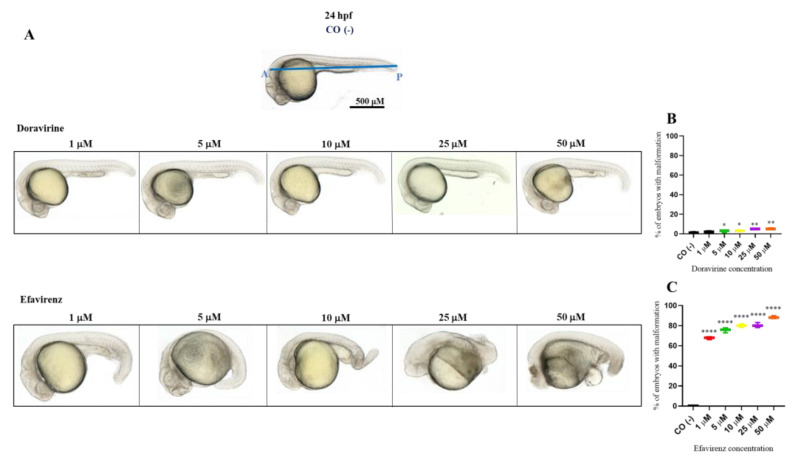
Representative pictures of the gross morphological effects of doravirine (DOR) end efavirenz (EFV) exposure during zebrafish embryo development. Dechorionated embryos were exposed with the immersion method to different DOR and EFV doses (1, 5, 10, 25, and 50 μM) from the gastrula stage (4 h post fertilization, hpf) up to 24 hpf for morphology observations. Control embryos [CO (-)] were treated with fish water plus 0.1% solvent (dimethyl sulfoxide, DMSO). (**A**) The pictures represent the morphological effects at 24 hpf. All pictures are lateral views with dorsal to the top and anterior to the left. (**B**,**C**) The graphs represent the percentages of embryos with malformations at 24 hpf. All treatments, including controls, were conducted 3 times, with 30 embryos for each experiment and each treatment. Embryos in the figures are representative of all experiments. The blue A–P line displayed in the control embryo pictures indicates the anterior–posterior (AP) axis. Results in the graphs are expressed as mean ± SD of three independent experiments, with 30 embryos for each experiment and each treatment. (* *p* < 0.05 vs. control group; ** *p* < 0.005 vs. control group; **** *p* < 0.0001 vs. control group).

**Figure 3 ijms-24-11664-f003:**
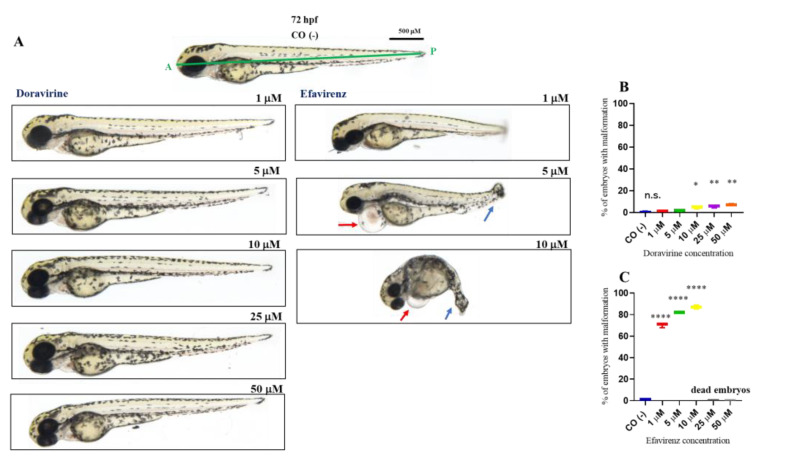
Representative pictures of the gross morphological effects of doravirine (DOR) end efavirenz (EFV) exposure during zebrafish embryo development. Dechorionated embryos were exposed with the immersion method to different DOR and EFV doses (1, 5, 10, 25, and 50 μM) from the gastrula stage (4 h post fertilization, hpf) up to 72 hpf for morphology observations. Control embryos [CO (-)] were treated with fish water plus 0.1% solvent (dimethyl sulfoxide, DMSO). (**A**) The pictures represent the morphological effects at 72 hpf. All pictures are lateral views with dorsal to the top and anterior to the left. (**B**,**C**) The graphs represent the percentages of embryos with malformations at 72 hpf. All treatments, including controls, were conducted 3 times, with 30 embryos for each experiment and each treatment. Embryos in the figures are representative of all experiments. The green A–P line displayed in the control embryo pictures indicates the anterior–posterior (AP) axis. The red arrows indicate the presence of pericardial edema, and the blue arrows indicate aberrant tail detachment and the necrosis process. Results in the graphs are expressed as mean ± SD of three independent experiments, with 30 embryos for each experiment and each treatment. (* *p* < 0.05 vs. control group; ** *p* < 0.005 vs. control group; **** *p* < 0.0001 vs. control group).

**Figure 4 ijms-24-11664-f004:**
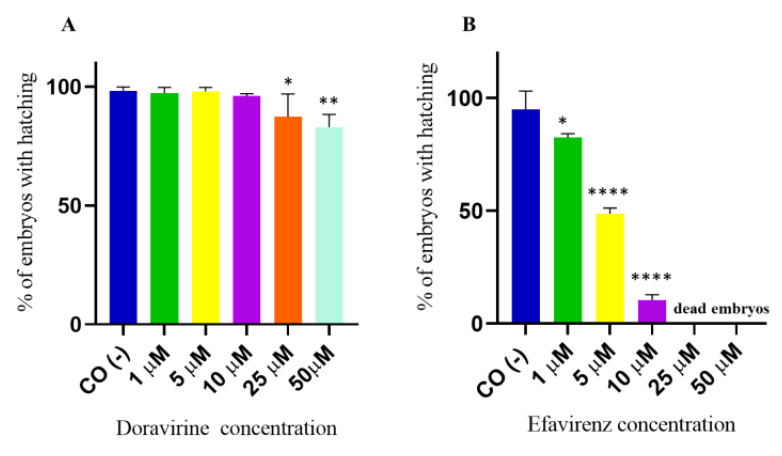
Percentages of hatched embryos at 72 h post fertilization (hpf) after exposure to (**A**) doravirine (DOR) and (**B**) efavirenz (EFV). Non-dechorionated embryos were exposed using the immersion method from the gastrula stage (4 hpf) and up to 72 hpf to different drug doses (1, 5, 10, 25, and 50 μM). Control embryos [CO (-)] were treated with fish water and 0.1% dimethyl sulfoxide (DMSO). The *X*-axis shows the drug doses used for the exposure of the embryos; the *Y*-axis shows the corresponding percentages of hatched embryos at 72 hpf. Results are expressed as mean ± SD of 3 independent experiments, with 30 embryos for each experiment and each treatment. (* *p* < 0.05 vs. control group; ** *p* < 0.005 vs. control group; **** *p* < 0.0001 vs. control group).

**Figure 5 ijms-24-11664-f005:**
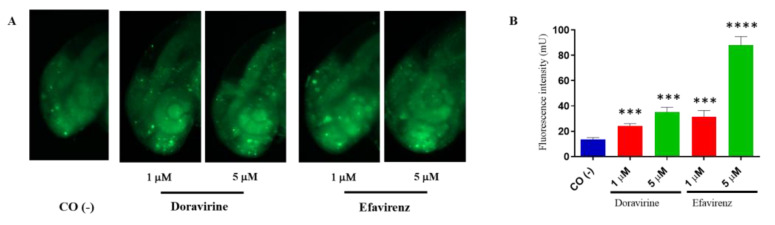
Doravirine (DOR) and efavirenz (EFV)-induced apoptosis in zebrafish embryo brains. After exposure with the drugs (1 and 5 μM doses for both drugs) from the gastrula stage (4 h post fertilization, hpf) up to 48 hpf, embryos were stained with acridine orange (AO). Control embryos [CO (-)] were treated with fish water and 0.1% dimethyl sulfoxide (DMSO) up to 48 hpf before AO staining. (**A**) Stained embryos were observed under a fluorescence microscope. All pictures are lateral views of the head and are representative of 3 independent experiments with 30 embryos for each experiment and each treatment. (**B**) Embryo heads were cut-off and homogenized, following which green fluorescence intensity was measured (pools of 20 embryos for each treatment). The *X*-axis shows the drug doses used for the exposure of the embryos; the *Y*-axis shows the measured fluorescence intensity (mU). Results are expressed as mean ± SD of three independent experiments. (*** *p* < 0.001 vs. control group; **** *p* < 0.001 vs. control group).

**Figure 6 ijms-24-11664-f006:**
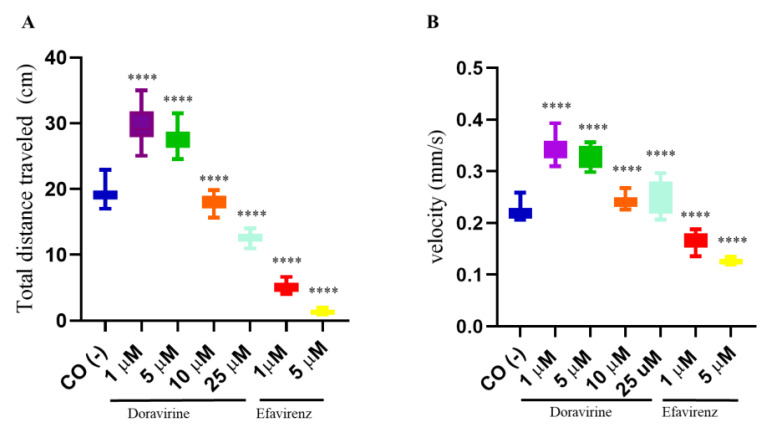
Swimming behavior of zebrafish larvae at 144 hpf analyzed with a light–dark locomotion test following treatments with doravirine (DOR) and efavirenz (EFV) using the immersion method. Embryos at 4 hpf were dechorionated and exposed until 144 hpf to drugs at the indicated concentrations (1, 5, 10, 25 μM for DOR; 1–5 μM for EFV) or fish water with 0.1% DMSO as the control [CO (-)] with the immersion method. (**A**) Movements were reported as mean ± SD total distance swam by the larvae (cm), calculated during both light and dark stimuli. (**B**) Speed (mm/s) was calculated during the same period. Results are expressed as mean ± SD of 3 independent experiments, with 12 survived larvae for each experiment and for each treatment. (**** *p* < 0.0001 vs. control group).

**Figure 7 ijms-24-11664-f007:**
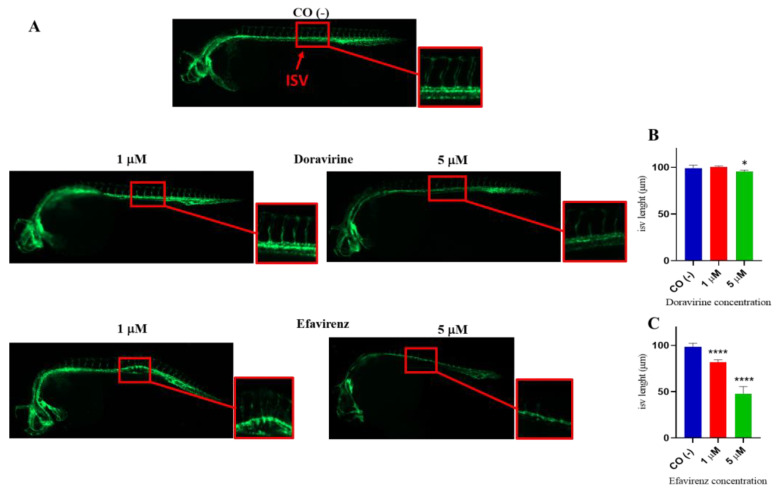
Effects of doravirine (DOR) and efavirenz (EFV) on angiogenesis and intersomitic vessel (ISV) development. Tg (*kdrl*: EGFP) embryos were exposed to drugs from the gastrula stage (4 h post fertilization, hpf) up to 30 hpf (1 and 5 μM doses for both drugs). Control embryos [CO (-)] were treated with fish water and 0.1% dimethyl sulfoxide (DMSO) up to 30 hpf. (**A**) Representative images of the control and treated embryos. Fluorescent vessels were observed under a fluorescence microscope. All pictures are lateral views with dorsal to the top and anterior to the left (magnification 20X). For all embryos, a magnified insert showing a zoomed view of the ISVs is also shown (magnification 32X) (**B**,**C**) The graphs represent the values obtained for the ISV length measurements both in the control and treated embryos. The *X*-axis shows the drug doses used for the exposure of the embryos; the *Y*-axis shows the measured length (μM). Results are expressed as mean ± SD of 3 independent experiments, with 30 embryos for each experiment and each treatment. For ISV length measurements, ten embryos per group were analyzed in each of the three experiments (* *p* < 0.05 vs. control group; **** *p* < 0.0001 vs. control group).

**Figure 8 ijms-24-11664-f008:**
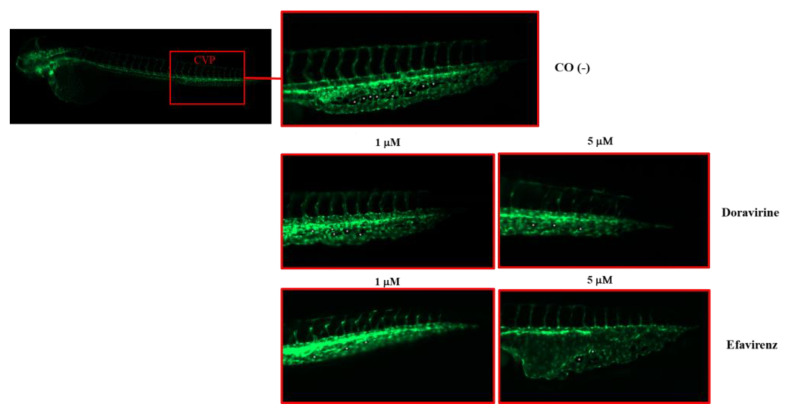
Effects of doravirine (DOR) and efavirenz (EFV) on caudal venous plexus (CSV) development. Tg (kdrl: EGFP) embryos were exposed to drugs from the gastrula stage (4 h post fertilization, hpf) up to 48 hpf (1 and 5 μM doses for both drugs). Control embryos [CO (-)] were treated with fish water and 0.1% dimethyl sulfoxide (DMSO) up to 48 hpf. Fluorescent vessels were observed under a fluorescence microscope. Images of the CVP of the control and treated embryos are representative of 3 independent experiments with 30 embryos for each experiment and each treatment. All pictures are lateral views with dorsal to the top and anterior to the left. For the control embryos, a picture representing the whole embryo is also shown (magnification 20×). For all embryos, a magnified insert showing a zoomed view of the CSV is shown (magnification 32×). Asterisks indicate intercapillary spaces. Enlargement of the CVP area with disruption of its morphology is evident following the 5 μM EFV treatment.

**Figure 9 ijms-24-11664-f009:**
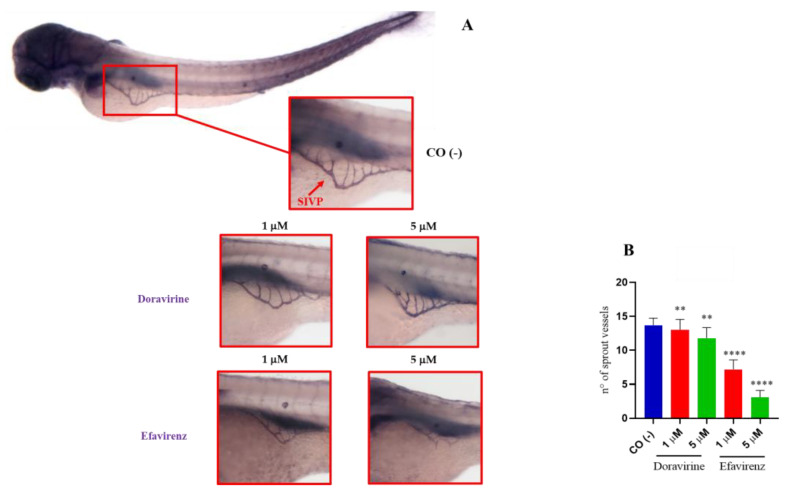
Effects of doravirine (DOR) and efavirenz (EFV) on the development of the sub-intestinal venous plexus (SIVP). Embryos were exposed to drugs from the gastrula stage (4 h post fertilization, hpf) up to 72 hpf (1 and 5 μM doses for both drugs). Control embryos [CO (-)] were treated with fish water and 0.1% dimethyl sulfoxide (DMSO) up to 72 hpf. Alkaline phosphatase (AP) staining was performed at 72 hpf to visualize the ectopic sprout formation of the SIVP in the embryos after treatments. (**A**) Representative image of a control embryo (lateral view with dorsal to the top and anterior to the left) (magnification 20X). Representative zoomed images of the control and treated embryos: the images of the stained SIVP were taken in lateral position at a 32× magnification. Images are representative of 3 independent experiments with 30 embryos for each experiment and each treatment. (**B**) The graph represents the average number of SIVP branches in the control and treated embryos. The *X*-axis shows drug doses used for the exposure of the embryos; the *Y*-axis shows the number of SIVP branches. Results are expressed as mean ± SD of 3 independent experiments, with 30 embryos for each experiment and each treatment. (** *p* < 0.005 vs. control group; **** *p* < 0.0001 vs. control group).

**Figure 10 ijms-24-11664-f010:**
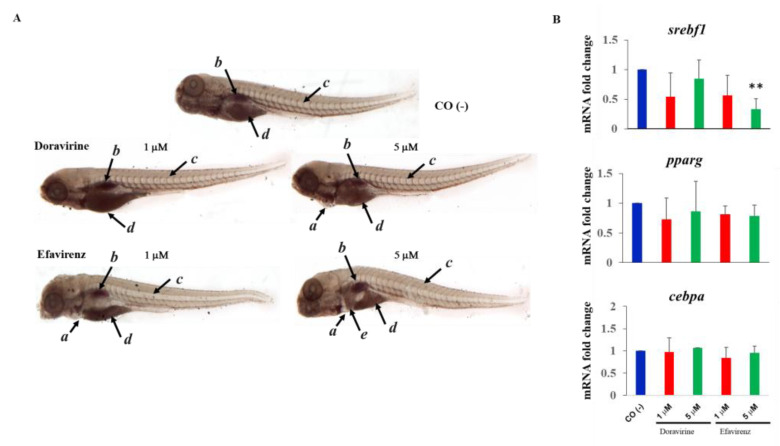
Effects of doravirine (DOR) and efavirenz (EFV) on lipid metabolism. (**A**) Effects of DOR and EFV on neutral lipid distribution. Embryos were exposed to drugs from the gastrula stage (4 h post fertilization, hpf) up to 144 hpf (1 and 5 μM doses for both drugs). Control embryos [CO (-)] were treated with fish water and 0.1% dimethyl sulfoxide (DMSO) up to 144 hpf. At 144 hpf, Oil Red O (ORO) staining was performed. Images of the control and treated embryos stained with ORO are representative of 3 independent experiments with 30 embryos for each experiment and each treatment. All pictures are lateral views with dorsal to the top and anterior to the left (magnification 20×). *a* pericardial edema, *b* lipid staining in the swim bladder, *c* lipid staining in the vasculature, *d* lipid staining of the residue yolk, and *e* lipid accumulation in the liver area. (**B**) Effects of DOR and EFV on *sterol regulatory element-binding transcription factor 1 (srebf1*), *peroxisome proliferator-activated receptor gamma* (*pparg*), and *CCAAT enhancer-binding protein alpha* (*cebpa*) gene expression. Embryos were exposed to drugs from the gastrula stage (4 h post fertilization, hpf) up to 48 hpf (1 and 5 μM doses for both drugs). Control embryos [CO (-)] were treated with fish water and 0.1% dimethyl sulfoxide (DMSO) up to 48 hpf. mRNA expression was measured using quantitative real-time RT-PCR, with the *ribosomal protein L13a* (*rpl13a*) having been used as housekeeping reference gene to normalize expression. mRNA expression is expressed as fold change with respect to the control embryos. The *X*-axis shows drug doses used for the exposure of the embryos; the *Y*-axis shows mRNA fold change. Results are expressed as mean ± SD of 3 independent experiments, with 30 embryos for each experiment and each treatment. (** *p* < 0.005 vs. control group).

**Table 1 ijms-24-11664-t001:** Criteria for grading (normal, mild, or severe phenotype) the treated embryos, according to their morphology at 24 and 72 hpf, respectively.

Morphological endPoints	Normal Phenotype	Mild Phenotype	Severe Phenotype
Head	Flexed	Straight	Extended
Anterior–posterior axis (AP)	Normal head and tail position	AP axis mildly disrupted	AP axis severely disrupted
Pericardial edema	Absent	Mild	Moderate to grave
Somites (24 hpf)	V-shaped	U-shaped	Severely disorganized or not formed
Tail	Properly detached and straight	Normal length but curved	Short length and curved, necrosis
Growth	Normal	Mild delay	Severe delay

**Table 2 ijms-24-11664-t002:** Summary of the findings obtained from zebrafish embryos after exposure to doravirine (DOR) and efavirenz (EFV) compared to the control embryos.

	Control	DOR	EFV
**Survival rate ***			
24 hpf	>90%	92–81%	80–44%
48 hpf	>85%	88–75%	72–0%
144 hpf	>85%	90–69%	not assessed
**Morphology ***			
24 hpf	0–1% mild abnormalities	0–5% mild abnormalities	68-88% severe abnormalities
72 hpf	0–1% mild abnormalities	1–7% mild abnormalities	70–100% severe abnormalities
**Hatching rate ***			
72 hpf	>98%	97–83%	83–0%
**Head apoptosis ****			
48 hpf	low	increased	highly increased (5 μM)
**Swimming behavior *****			
144 hpf	normal swimming	movement activation up to 5 μM	movement reduction up to 5 μM
		movement reduction above 5 μM	not assessed above 5 μM
**Angiogenesis ****			
*Kdrl*:EGFP 30 hpf	normal	slightly lower only at 5 μM	very low at both doses
ISV 30 hpf	proper morphology/length	proper morphology/length	deformed/shorter at both doses
CVP 48 hpf	proper morphology	reduced intercapillary spaces (5 μM)	reduced intercapillary spaces (1 μM)
			enlarged area (5 μM)
SIVP 72 hpf	proper morphology	slightly smaller basket (5 μM)	smaller basket (1 μM)
			very small or no basket (5 μM)
**Neutral lipid distribution ****			
144 hpf	proper	proper	lipid storage in the liver (5 μM)

Notes: * (1–5–10–25–50 μM) = doses used for doravirine (DOR) and efavirenz (EFV) exposure; ** (1–5 μM) = doses used for doravirine (DOR) and efavirenz (EFV) exposure; *** (1–5–10–25 μM) = doses used for doravirine (DOR) and efavirenz (EFV) exposure.

## Data Availability

Data are available upon request; the interested researchers could contact directly Eugenia Quiros-Roldan (eugeniaquiros@yahoo.it), sabella Zanella (isabella.zanella@unibs.it), Daniela Zizioli (daniela.zizioli@unibs.it).

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
