# Peer review of "Comparison of Efavirenz and Doravirine Developmental Toxicity in an Embryo Animal Model"

_ijms, 2023, doi:10.3390/ijms241411664_

Round 1
Reviewer 1 Report
In the present manuscript, titled “Comparison of efavirenz and doravirine developmental toxicity in an embryo animal model”, Zizioli and co-workers investigated the developmental toxicity of doravirine (DOR), compared to efavirenz (EFV), using zebrafish embryos as an in vivo model.
They examined survival, morphology, hatching rate, apoptosis in the developing head, locomotion behavior, vasculature development and neutral lipid distribution, highlighting that DOR showed a better safety profile than EFV, with a very low mortality rate and mild morphological alterations. Therefore, I would like to recommend the submitted article for publication after addressing the following comments and details:
1. Nothing about in vitro/in vivo studies already reported in literature has been discussed in introduction. It is my opinion that these aspects should be significantly improved with an additional comparable analysis between the previous works and your study, discussing the novelty of the paper. Furthermore, 2D structures of efavirenz and doravirine must be showed to appreciate structural differences between drugs.
2. In the Discussion paragraph, a summary table, covering all the data obtained from the study, must be included. In particular, the table must present data after the treatment with efavirenz and doravirine and data without treatment to analyze differences.
3. Authors need to discuss the merits and limitations of the study.
Minor editing of English language required
Author Response
Please, see the attached file.

Reviewer 2 Report
This is a study comparing efavirenz and doravirine toxicity in zebrafish embryos. The strength of the study is the extensive experimentation involved. One of the major comments that I have is for the authors to emphasize that this is an animal study and that further human studies/post-marketing studies and surveillance registries are needed to confirm these results. My specific comments/inquiries are as follows:
1. In the abstract and the text, please clarify how many zebrafish embryos were studied in each arm (sample size). In Lines 30 and 31, "DOR induced a very low mortality rate and mild morphological alterations compared to EFV exposure also in the sub-therapeutic ranges. Further, DOR only slightly affected hatching rate at supra-therapeutic doses while EFV strongly reduced hatching already at sub-therapeutic doses." Please indicate the mortality and hatching rates in both arms within a specified drug concentration.
2. Abstract conclusion: Line 39, indicate that further human studies are needed to confirm these results from animal experiments
3. Throughout the manuscript, please change "HIV-infected" into 'people with HIV.' This is now the preferred terminology to lessen stigma.
4. Please clarify in the methods whether the laboratory assessments were performed by blinded laboratory personnel. That is, were the people performing the measurements aware OR unaware whether the zebrafish received doravirine or efavirenz? This needs to be stated in the Methods.
5. Line 118: "small percentage" -- please state the actual percentage here. Similar to Line 168 ("slight but significant reduction of hatching rate") and Line 365 ("in a little percentage of embryos"). Throughout the manuscript, please state percentages and rates when available.
6. Discussion: The statement "Our results showed that DOR does not affect embryogenesis" is NOT accurate. Rephrase this statement into "DOR had minimal effects" since there were decreases in survival rates in DOR.
7. Line 505: revise for clarity
8. Line 570: DOR... "without evident or lethal teratogenic effects" is not supported by the data. Please rephrase this statement.
9. Conclusion: Indicate that these findings are in animal studies and further surveillance and post-marketing human studies are warranted to confirm their translation in humans.
10. The authors need to expound more on the side effects of doravirine in human studies. The way this paper was written appears to be off-balanced, citing multiple toxicities in efavirenz but only glossing over doravirine's side effects stating "very few studies on DOR safety has been published". It must be noted that in Reference 26 (Gatell, 2019), "CNS AEs were reported by 26.9% and 47.2% of doravirine and efavirenz recipients, respectively." Although doravirine had significantly lower CNS effects, this was still reported by 1 in 4 patients. Please expound on the published doravirine AEs in the Discussion.
11. How was doravirine and efavirenz purchased? Were these provided for free by the manufacturers? If so, this has to be declared. If medical writers and other funding were utilized, this also must be declared.
12. Competing Interests: Dr. Quiros-Roldan previously declared "E.Q.R. received travel grants from Bristol-Myers Squibb, Gilead Sciences, ViiV Healthcare, Janssen-Cilag, Merck Sharp & Dohme and consultancy fees from Janssen-Cilag, ViiV Healthcare and Merck Sharp & Dohme." This conflict of interest was found in the paper " The possible mechanisms of action of 4-aminoquinolines (chloroquine/hydroxychloroquine) against Sars-Cov-2 infection (COVID-19): A role for iron homeostasis?"
Pifeltro is manufactured my Merck. Is there a reason why this was not disclosed in this paper? Please confirm whether all other authors have no competing interest.
"Notwithstanding" was used multiple times and suggest to change this in certain areas.
Discussion, Line 405: rephrase the second paragraph for clarity. Specifically, it was unclear what "difficult to overcome" meant.
Author Response
Please, see the attached file.

Round 2
Reviewer 1 Report
All suggestions were well followed
Reviewer 2 Report
The authors have adequately addressed my comments/inquiries. I believe that the paper can now proceed with publication after minor English language editing/proofreading. For example, in the Abstract, all percentages should have a '%' sign: ([survival rates 92, 90, 88, 88 and 81%] should be [survival rates: 92%, 90%, 88%, 88% and 81%]...
Minor English language editing/proofreading needed.